# Biomass Characteristics and Their Effect on Membrane Bioreactor Fouling

**DOI:** 10.3390/molecules24162867

**Published:** 2019-08-07

**Authors:** Petros K. Gkotsis, Anastasios I. Zouboulis

**Affiliations:** Laboratory of Chemical and Environmental Technology, Section of Chemical Technology and Industrial Chemistry, School of Chemistry, Aristotle University of Thessaloniki, 54124 Thessaloniki, Greece

**Keywords:** membrane bio-rectors, fouling, biomass characteristics, EPS, SMP, mixed liquor characterization

## Abstract

Biomass characteristics are regarded as particularly influential for fouling in Membrane Bio-Reactors (MBRs). They primarily include the Mixed Liquor Suspended Solids (MLSS), the colloids and the Extracellular Polymeric Substances (EPS). Among them, the soluble part of EPS, which is also known as Soluble Microbial Products (SMP), is the most significant foulant, i.e., it is principally responsible for membrane fouling and affects all fundamental fouling indices, such as the Trans-Membrane Pressure (TMP) and the membrane resistance and permeability. Recent research in the field of MBRs, tends to consider the carbohydrate fraction of SMP (SMP_c_) the most important characteristic for fouling, mainly due to the hydrophilic and gelling properties, which are exhibited by polysaccharides and allow them to be easily attached on the membrane surface. Other wastewater and biomass characteristics, which affect indirectly membrane fouling, include temperature, viscosity, dissolved oxygen (DO), foaming, hydrophobicity and surface charge. The main methods employed for the characterization and assessment of biomass quality, in terms of filterability and fouling potential, can be divided into direct (such as FDT, SFI, TTF_100_, MFI, DFCM) or indirect (such as CST, TOC, PSA, RH) methods, and they are shortly presented in this review.

## 1. Introduction

Membrane fouling, which is defined as the undesirable deposition of particle, colloidal and dissolved matter onto the membrane surface or within the membrane pores, is still the major disadvantage of the state-of-the-art Membrane Bio-Reactor (MBR) technology. Although the MBR technology has made significant progress over the last few years finding new application fields, especially for difficult to be treated industrial wastewaters, membrane fouling still remains the primary drawback for its further applicability, since it reduces the system’s productivity, increases the energy requirement, especially for gas scouring and frequency of chemical cleaning, and results in frequent membrane replacement [1,2]. As a result, there has been a rapid increase of the scientific publications, which focus on the study of MBR systems and particularly on the membrane fouling issues (Figure 1).

Several membrane fouling mechanisms have been proposed, including pore clogging, gel layer formation, cake layer formation and osmotic pressure effect (Figure 2). During the complete pore clogging, it is assumed that each particle reaching the membrane blocks a pore without superimposing over other particles. During the gel layer formation, the depositing particles do not block the membrane pores, either because the membrane is dense and there are no pores to block, or because the pores are already covered by other particles. Although in some cases only a gel layer formation was observed during long term MBR operation, this can be regarded as a special form of cake layer, which is characterized by the accumulation of more foulants on the membrane surface. Finally, osmotic pressure effect refers to the osmotic gradient which is induced by the significantly higher concentration of ions in the cake layer, compared to that in the permeate side [3]. Recently, the study of Tang et al. [4] provided a unified thermodynamic mechanism for fouling behaviour, based on energy balance principles, and significantly improved the fundamental understanding of fouling.

Among the factors which influence fouling, i.e., the wastewater characteristics, the applied operating conditions, the membrane properties and the biomass characteristics, the latter are considered to be the most important and, thus, have been the subject of numerous research studies in the field of environmental technology and wastewater treatment using MBRs. The most common classification of biomass characteristics is based on the physico-chemical nature of contained substances and more specifically on their size, which was initially considered to be crucial for membrane fouling. Hence, biomass characteristics can be classified into three major categories: suspended solids (SS), colloids and soluble matter. These are regarded as potential foulants, i.e., substances which are responsible for membrane fouling. Initially the concentration of suspended solids was (incorrectly) thought to cause severe membrane fouling, however, later the focus quickly turned to the soluble sticky substances which were found to be either bound to the microbial aggregates (bio-flocs), or freely suspended in the mixed liquor. These substances are referred to as Extracellular Polymeric Substances (EPS) and include a wide range of macromolecules, such as polysaccharides, proteins, nucleic acids, phospholipids and other polymeric compounds, which are usually at/or outside the microbial cell surface and in the intercellular space of bio-flocs as well.

In most cases, polysaccharides and proteins are assumed to be the most significant foulants, especially when they are in their soluble form, which is also known as Soluble Microbial Products (SMP). Thus, the determination of EPS and SMP concentrations relies mostly on the measurement of polysaccharides and proteins. Other wastewater and biomass properties usually include temperature, viscosity, dissolved oxygen (DO), foaming, hydrophobicity and surface charge. However, these characteristics affect membrane fouling mainly indirectly, through the changes they induce to the major characteristics, such as MLSS, colloidal matter, EPS, and to the system’s biology.

Regarding the methods which are employed in order to assess the mixed liquor’s filterability and the membrane fouling potential, these can be either direct, when they are based on free drainage, vacuum drainage and cross-flow filtration, or indirect, when they are based on the measurement/analysis of Capillary Suction Time (CST), colloidal Total Organic Carbon (TOC), particle size and relative hydrophobicity (RH). Nevertheless, although more than one of these methods may be useful for a given case under examination, none of them is universally acceptable for all the practically found conditions.

The present paper attempts to provide an integrated review of the biomass characteristics in Membrane Bio-Reactors (MBRs) and their effect on membrane fouling. Both the fundamental and the secondary biomass properties are presented and their relationship with membrane permeability loss and fouling is thoroughly discussed. Additionally, the most commonly employed biomass characterization methods examined in relevant treatment systems are shortly presented, in terms of membrane permeability and fouling potential.

## 2. Major Biomass Characteristics Which Affect Membrane Fouling

### 2.1. Mixed Liquor Suspended Solids (MLSS)

#### 2.1.1. Misinterpreted Role of MLSS in MBR Systems

In the early days of MBR technology, the MLSS concentration was regarded as the primary factor, which causes membrane fouling. On the basis of this assumption, it was considered that the performance of these systems could be improved by lowering the MLSS concentration in the membrane tank. For this reason, in some studies, specially designed sedimentation tanks were placed between the aeration tank and the membrane tank in order to partially or substantially reduce the MLSS concentration before the mixed liquor entered the (following) membrane tank. However, this hypothesis turned out to be incorrect based on numerous observations, which were made during the operation of various lab- and pilot-scale MBRs and showed that the impact of MLSS concentration on membrane permeability can be occasionally negative, positive or even insignificant. More specifically, Bin et al. [5] observed that permeate flux decreased with increasing the MLSS concentration and attributed it to the rapid formation of a fouling cake layer at high MLSS concentrations. On the contrary, at low MLSS concentrations the progressive pore blocking, which was created by the presence of colloids and soluble substances, delayed the fouling of membrane.

Chang and Kim [6] compared the operation of two MBRs with and without a sedimentation tank between the aeration tank and the membrane tank. The MLSS concentration of the pre-settled and the unsettled mixed liquor, which entered the membrane tank was 100 mg/L and 6000 mg/L, respectively. Submerged hollow fibre membranes, made of polysulfone, were used for both systems, while scouring air was supplied from the bottom of the membrane bundle. Contrary to expectations, higher permeability loss was observed for the low MLSS system. Lee et al. [7] performed side-by-side experiments using two MBRs with suspended or attached biomass in order to investigate the effect of MLSS concentration. The MLSS of suspended biomass MBR was approximately 3000 mg/L, while that of attached biomass MBR was only around 100 mg/L. It was observed that the average particle size was smaller and the increase rate of Trans-Membrane Pressure (TMP) was seven times higher for the low MLSS concentration (i.e., for the attached biomass MBR). However, Scanning Electron Microscopy (SEM) measurements showed that the cake layer was much thinner, than that with the higher MLSS concentration (i.e., for the suspended biomass MBR). This observation suggested that the cake layer is much denser, when low MLSS concentrations are employed. Rosenberger et al. [8] showed that the increase of MLSS concentration reduced fouling at low concentrations (~6 g/L) and increased fouling at high concentrations (>15 g/L). Meanwhile, the level of MLSS concentration did not appear to have a significant effect on membrane fouling between 8 and 12 g/L. Similarly, Hong et al. [9] found that flux was not affected by the MLSS concentration within a moderate range of concentrations (~3.5–8.5 g/L). All the aforementioned observations suggest that the MLSS concentration alone cannot be used as a reliable fouling index and must be always examined taking into account also other system parameters as well, such as the operating conditions, wastewater and membrane characteristics etc. [10,11].

#### 2.1.2. Efforts to Correlate MLSS with Fouling-Critical Values

Although various empirical relationships have been suggested for the prediction of flux from the MLSS concentration, their application is limited, because they are obtained under very specific conditions. In addition, they are based on a limited number of operating parameters, while other parameters are neglected. Cho et al. [12] proposed a mathematical expression which links the MLSS concentration, EPS and TMP with the specific cake resistance. In their study, the specific cake resistance was slightly changed, when the MLSS concentration was between 4 and 10 g/L at constant EPS and TMP values. The lack of a clear correlation between the MLSS concentration and any specific fouling characteristic indicate that this is a poor indicator of the MBR fouling propensity. More recent studies have also pointed the presence of colloidal and dissolved organic matter, rather than the MLSS concentration, as being the primary foulants in MBRs; however, a relationship between fouling and biomass characteristics or operating parameters still remains a challenging issue.

Nevertheless, although the MLSS concentration is now considered to be weakly correlated with membrane fouling in MBRs, extreme values of this parameter should be avoided, because they can cause accelerated fouling. Lubbecke et al. [13] suggested the existence of a threshold value above which the MLSS concentration has a fully negative influence on fouling (30 g/L). Excessively high MLSS concentrations can dramatically deteriorate the air scouring efficiency by hampering the bubbles’ movement and the membrane fibre vibration, due to the increased mixed liquor viscosity. Moreover, high viscosity dampens the back-transport effect and increases the net force towards the membrane surface, leading to the deposition of sludge flocs, small particles and bio-molecules. High MLSS concentrations can lead also to low Oxygen Transfer Efficiency (OTE), reduce the DO concentration and increase membrane fouling. Additionally, extremely high MLSS concentrations promote the formation of ‘dead zones’ in the mixed liquor, without sufficient mixing.

On the contrary, excessively low MLSS concentrations can reduce the Solids Retention Time (SRT) and increase the Food to Microorganisms (F/M) ratio, which in turn may promote an accelerated membrane fouling. Current studies have shown that a concentration of 10 g/L can be a critical point with respect to membrane fouling [10,11,14]. Similarly, Wu and Huang [15] suggested that MLSS concentrations higher than 10 g/L increase sludge viscosity resulting in poor filterability. Lousada-Ferreira et al. [16] conducted cross-flow filtration experiments by employing various MLSS concentrations (3.6–18.3 g/L). The respective results showed that at higher MLSS concentrations (>10 g/L), smaller particles were adsorbed (<20 μm) and the system exhibited lower filterability. Finally, it should not be omitted that the MLSS concentration also impacts on the MBR removal efficiency; a concentration of ~6 g/L has been identified as optimum, based on the removal of COD [17].

#### 2.1.3. The Weak Correlation between MLSS Concentration and Membrane Fouling

The weak correlation between the MLSS concentration and membrane fouling can be explained by the critical flux theory. According to this theory, primarily the submicron-sized, fine particles and macromolecules deposit on the membrane surface, whereas the larger bio-flocs are transported away, because of the particle back-transport phenomenon. Since fine particles and macromolecules comprise only the smaller fraction of MLSS, the MLSS concentration itself cannot be strongly correlated with membrane fouling. However, as aforementioned, extremely high or low MLSS concentrations can cause unfavourable conditions for membrane filtration, such as slow mixing, low DO concentration, ‘dead zone’ formation, SRT reduction etc., especially if they are combined with poor system design and operating conditions.

The effect of MLSS concentration on membrane fouling can be also erroneously interpreted, because of the complex interactions among the operational parameters. This is especially true, when the operating parameters are overlooked or not controlled properly. Some of the common reasons which can lead to wrong conclusions, regarding the effect of MLSS on fouling, are the following:In a hypothetical side-by-side experiment, the same operating conditions are applied in two reactors, including the feed flow rate, the aeration rate and the flux, except for the MLSS concentration. Under this condition, the reactor with the lower MLSS concentration has a lower SRT and a higher F/M ratio, than its counterpart, resulting in increased fouling. Although low SRT and high F/M ratio are the main causes of higher fouling rate, it may be concluded that membrane fouling is more significant at lower MLSS concentrations.In another hypothetical example, the MLSS concentration can be gradually increased in order to study its effect on membrane fouling. Excess sludge removal may be paused or reduced to enable the increase of MLSS in this case. For these (dynamic) conditions, the apparent SRT, which is calculated by dividing the total sludge volume in the reactor by the sludge, which is removed every day, is no longer the effective SRT. Meanwhile, the F/M ratio is dynamically decreased, due to the increase of MLSS over time. Since these two crucial, fouling-affecting factors are dynamically changing, any observation made during this condition can be considered as irrelevant to the influence of MLSS. If the DO concentration and the mixing pattern also change as the MLSS concentration increases, then the observations made during the experiment cannot be regarded as valid.Conversely, if the MLSS concentration is gradually decreased by increasing the biomass removal, the effect of SRT on the lower MLSS concentration is higher, than the effect of steady-state SRT on MLSS. If the membrane fouling rate is measured under this condition, the apparent conclusion is that fouling increases as the MLSS concentration decreases.As the MLSS concentration increases, the OTE tends to decrease, because of the physical hindrance of the solids against the oxygen transfer to liquid, in the gas-liquid interface. As a consequence, the DO concentration can decrease, which in turn causes non-ideal conditions for the microorganisms and promote membrane fouling. If the effect of MLSS on DO is overlooked, then the higher MLSS concentration seems to be the cause of high fouling rate [11].

#### 2.1.4. Bio-Floc Size

Given the large size of solids compared to the membrane pore size, pore blocking by the bio-flocs themselves is not quite possible. However, they can still contribute to fouling through the (secondary) production of soluble substances. In addition, it has been reported that membrane permeability is positively correlated with the size of bio-flocs. From a hydrodynamic standpoint, larger flocs could be dragged away from the membrane by high shear-induced diffusion and by inertial forces, as well as by low Brownian diffusion. The deposition of larger and looser sludge flocs encourages the formation of a more porous and permeable cake layer, resulting in the reduction of fouling resistance. On the other hand, Brownian diffusion controls the motion of smaller flocs at lower shear stress. The deposition of smaller flocs on the membrane surface may result in the formation of a less porous cake layer with smaller pore size, which in turn can increase the hydraulic cake resistance [14].

### 2.2. Colloidal Matter

Colloids can be defined as the fine particles with a size ranged between 0.001–1 μm (1–1000 nm). Colloids can directly cause fouling mainly through pore blocking, when each particle blocks a pore without superimposing over other particles, or standard blocking, when the particles deposit within the membrane pore and the pore volume decreases proportionally to the volume of deposited particles. Nevertheless, they also contribute to the formation of the cake layer on the membrane surface; it has been reported that almost 90% of the particles in the cake layer are smaller than 0.3 μm. Colloidal matter can be classified into organic macromolecules and rigid inorganic colloids. The former can be further classified into bio-molecules (mainly polysaccharides and proteins) and fulvic compounds, while the latter may include silica, aluminum silicate minerals and iron(oxy) hydroxides in aerobic MBRs, or struvite and calcium carbonate in anaerobic MBRs. It is generally accepted that colloids in MBRs derive from the influent substrate or through the microbial metabolism [3].

### 2.3. Extracellular Polymeric Substances (EPS)

#### 2.3.1. Definition and Classification

Extracellular Polymeric Substances (EPS) have received considerable attention recently, since membrane fouling in MBRs has been largely attributed to these compounds. EPS is a general term, encompassing all classes of macromolecules, such as polysaccharides, proteins, nucleic acids, phospholipids and other polymeric compounds, which are found at, or outside the microbial cell surface, as well as in the intercellular space of bio-flocs (Figure 3). The fundamental functions of EPS include the aggregation of bacterial cells in bio-flocs and bio-films, the formation of a protective barrier around the bacteria, the retention of water and the adhesion to surfaces. With its heterogeneous and changing nature, EPS can form a highly hydrated gel matrix, in which microbial cells are embedded and, thus, create a significant barrier to permeate flow in membrane processes. Finally, the bio-flocs, which are attached on the membrane, can provide a major nutrient source during the formation of bio-film and also enable microorganisms to assimilate trace metals that are required for their metabolism [18].

EPS are classified into those which derive directly from the microbial cell wall, also known as bound-EPS (bEPS), and those which are not associated with the microbial cell, but are solubilized in the mixed liquor, also known as soluble-EPS (sEPS). Bound-EPS are further divided into tightly bound EPS (tb-EPS), which are strongly attached onto the microbial or bio-floc surface, and loosely bound EPS (lb-EPS) which can be easily detached from them. Soluble EPS are also reported in literature as Soluble Microbial Products (SMP) and mainly consist of polysaccharides, proteins, nucleic acids, enzymes and lipids. SMP have many different origins, e.g., they can be part of the debris of ruptured/hydrolysed cells during the cell lysis process, directly secreted by the microorganisms, or even provided by the feed substrate. SMP are generally classified into two categories: Utilization Associated Products (UAP), which derive from the biodegradation of original substrates during the microbial growth phase, and Biomass-Associated Products (BAP), which are released during biomass decay in the endogenous phase [19].

Other substances of interest in the MBR field include the Transparent Exopolymer Particles (TEP) and the Bio-Polymer Clusters (BPC). Originally thought to play a major role in the bio-films formed on reverse osmosis (RO) membranes, TEP have also been extensively studied in MBR processes. TEP mainly consist of polysaccharides, but they can also include proteins, lipids, nucleic acids etc. They are usually transparent sticky gel particles and are largely overlapped with SMP in nature. TEP are known to be formed by long-chain polysaccharide backbones, which trap other substances of smaller size, and their size ranges from nanometre to millimetre. In some cases, TEP concentration in the supernatant can be used as an index of the membrane fouling potential. BPC have been reported to act as ‘glue’ and, thus, promote the deposition of biomass on the membrane surface, which in turn accelerates the formation of a sticky and impermeable cake layer. Nevertheless, BPC can be rather easily detached from the membrane surface by aeration using coarse bubbles and diffused into the mixed liquor [11,14].

#### 2.3.2. The Dominant Role of SMP

As previously stated, the MLSS were considered to be a major foulant in the early years of MBR. However, the focus moved later to EPS presence after finding out that it was the bio-molecules (polysaccharides, proteins etc.), which may contribute mostly to cake layer formation on the membrane surface during typical operating conditions. Additionally, it was observed that it was the EPS content of the mixed liquor that was more closely related with the microbial physiology, than the MLSS. Since then, numerous studies have confirmed the positive correlation of EPS concentration with the membrane fouling rate. Nevertheless, despite the influence of EPS on the biological activity, and especially on the bio-film formation, recent research tends to focus more on the effect of its soluble fraction (SMP) on fouling. When compared to other biomass characteristics (e.g., MLSS, particle size, bound-EPS etc.), the SMP have the strongest relationship with membrane fouling rate, usually expressed as dTMP/dt. During filtration, the SMP are thought to adsorb onto the membrane surface, block the membranes pores and/or form a gel layer on the surface, where they provide a possible nutrient source for bio-film formation and increase the resistance to permeate flow. More specifically, the larger molecular-weight fractions of the SMP (i.e., MW > 100 kDa), which consist mainly of polysaccharides and proteins, can cause severe pore blocking and flux decline [11,14].

However, the fouling rate cannot be simply correlated with the quantity of SMP. The complex relationship between the SMP concentration and fouling rate is not surprising, since the notion started simply because the SMP were found to be the major constituents of fouling layer on the membrane surface. It should be noticed that this hypothesis is based on the dubious assumption that the SMP, which may be produced under different conditions, have an equal tendency to foul the membrane. Nonetheless, the fuzzy relationship between the SMP concentration and the resulting fouling has disproven this hypothesis. Another possible reason is the fact that the analytical methods, which are used to quantify the SMP fraction do not detect accurately all the biomolecules (polysaccharides, proteins, etc.). Furthermore, the actual membrane foulants are not necessarily detected by the analytical methods, which are used to quantify the SMP; apart from polysaccharides and proteins, which are considered to be the major foulants, the available analytical methods can also detect terrestrial humic substances, non-biological polymeric substances etc. The lack of a direct relationship between the biological parameters measured in the reactor and the extent of MBR fouling is also due to the preferential deposition of materials onto the membrane surface. Characterization measurements have shown that the compositions of cake layer and of the bulk can differ significantly, with the concentrations of polysaccharides and proteins being higher on the membrane surface. In addition, non-uniformity within any cake layer has been also observed [10,11].

#### 2.3.3. Carbohydrate and Protein Fraction

EPS (bound or soluble) are usually quantified as carbohydrate and protein equivalents, since the majority of them consists of these two bio-molecule categories. Therefore, EPS are typically characterized and quantified according to their relative content of carbohydrate (EPS_c_) and protein (EPS_p_) (see also Section 2.3.5). The EPS_p_ concentration is usually higher than the EPS_c_ concentration, while typical concentration values range between 11–120 mg/L and 7–40 mg/L, respectively.

Although the SMP_p_ play an important role in fouling, as implied by the significant amount of proteins, which is retained by the membrane (15–90%), their correlation with fouling has been less widely reported. In most cases, the concentration of the carbohydrate fraction (SMP_c_) appears to be the major contributor to membrane fouling, as compared to the protein one (SMP_p_). The statement that SMP_c_ are more significant than SMP_p_ is reinforced by the results from the respective analysis of the cake layer, which have revealed higher carbohydrate and lower protein concentrations, when compared to those in the mixed liquor. In addition, the high MW compounds (>100 kDa), both in the mixed liquor supernatant and in the cake layer, are mainly polysaccharides.

The high fouling propensity of SMP_c_ can be attributed to the hydrophilic nature of polysaccharides, as compared to SMP_p_, which are generally regarded as hydrophobic. The strong interaction between the commonly used MBR membranes, which are generally hydrophilic, and the hydrophilic organic compounds is usually the cause of initial fouling. However, the nature of SMP_c_ can change during the unsteady operation of an MBR system, and thus, they cannot be easily correlated to fouling. SMP_c_ strongly contribute to membrane fouling due also to the gelling properties of polysaccharides [20]. The latter can be attributed to the network, which is created by their cross-linked chains. Polysaccharides are composed of numerous blocks (monomers), such as α-guluronic acid and β-mannuronic acid, which interact with each other and enable the formation of cross-links. Particularly, the gelation can be enhanced in the presence of divalent or multi-valent cations, because they act as bridges for the carboxyl groups of the polysaccharides. Consequently, the presence of such cations can lead to the formation of an impermeable gel layer on the membrane surface, which significantly increases the filtration resistance, and also enables more bacteria to be attached, either by a bridging bacteria-membrane mechanism, or by providing a suitable substrate for bacteria [2,10].

Finally, it must be stressed that high polysaccharide and protein concentrations do not necessarily accelerate membrane fouling. Many operating parameters, which affect the SMP concentration, such as HRT, SRT, F/M ratio etc., should be also taken into consideration and it is very unlikely that the concentration of the SMP_c_ or SMP_p_ alone could predict the fouling propensity [11].

#### 2.3.4. Contribution of Humic Substances

Although their measurement is generally overlooked for polysaccharides and proteins, humic substances are found in significant concentrations in the mixed liquor of MBR systems. Humics tend to be adsorbed on the membrane surface; however, apart from the direct contact with the -usually hydrophilic- membrane, they tend to interact with other compounds as well, due to their strong hydrophobic nature. This facilitates the deposition of more bio-molecules (proteins, polysaccharides etc.), resulting in the gradual increase of fouling [2].

#### 2.3.5. Extraction and Quantification of EPS

In general, no standard method for the direct quantification of EPS exists and their analysis relies on their preliminary extraction from the mixed liquor bio-flocs. The accurate quantification of EPS is not easy, because of the difficulties which are encountered during their separation from the rest of mixed liquor constituents and are based on the various chemical properties of bio-molecules with different structures. Therefore, there are no standardized quantification methods and the measured EPS concentration can vary widely even in the same mixed liquor sample, depending on the employed experimental method. As a result, direct comparisons of respective data obtained from different studies are not generally valid [21].

To quantify the EPS content, the mixed liquor sample is centrifuged and the supernatant is obtained. Alternatively, the sample can be filtered and the permeate is used to measure the EPS concentration. It is understood that the EPS, which is measured in the supernatant or permeate, refers to the soluble part, i.e., the SMP. The carbohydrate (SMP_c_) and protein (SMP_p_) fractions are usually quantified with the Dubois method [22] and the Lowry method [23], respectively. It must be also stated that the quantified EPS do not represent actual masses, but glucose and bovine serum albumin (BSA) equivalent masses, since the respective calibrations are based on glucose and BSA solutions, respectively. However, EPS can be also analysed by means of more sophisticated analytical techniques, which enable the identification of specific molecules, such as High-Performance Size Exclusion Chromatography (HPSEC), which is widely used for the analysis of potable raw water, Fourier Transform Infra-Red (FTIR), Nuclear Magnetic Resonance (NMR) spectroscopy and Liquid Chromatography-Organic Carbon and Nitrogen Detector (LC-OCND), which combines the advantages of HPSEC and TOC (Total Organic Carbon) systems. All these techniques have been successfully used to characterize extensively the EPS and the other organic compounds in the mixed liquor of various MBR systems [10,11].

## 3. Other Fouling-Related Wastewater and Biomass Characteristics

### 3.1. Temperature

Temperature affects membrane filtration through permeate viscosity. The following Equation (1) is usually applied for the temperature correction:J = J_20_ 1.025^(T−20)^(1)
where J is the flux at the process temperature T and J_20_ is the flux at 20 °C. It has been observed that the normalized resistance at lower temperatures is greater than expected. This is explained by a number of contributing phenomena, such as:The increase of sludge viscosity, which is more significant than the permeate viscosity and can reduce the shear stress generated by the coarse air bubbling, commonly applied for membrane air scouring cleaning purposes.The intense deflocculation at low temperatures, which reduces the bio-floc size and releases EPS.The particle back-transport velocity, which decreases linearly with temperature increase, according to Brownian diffusion.The biodegradation of COD, which decreases with temperature increase and results in the high concentration of unbiodegraded soluble and particulate COD.

All these factors directly impact on membrane fouling and, as a result an increase of foulants deposition is expected to take place on the membrane surface at lower temperatures [10].

### 3.2. Viscosity

Viscosity is closely related to the mixed liquor temperature and the concentration of suspended solids. While viscosity has been reported to increase exponentially with the MLSS concentration, there is a critical MLSS value below which viscosity remains low, and rises slowly with the increase of MLSS, and above which it increases exponentially with the increase of MLSS. This critical value generally ranges between 10–17 g/L, depending on feedwater quality and operating conditions. Viscosity impacts both on the flux and on the air, which is employed for membrane scouring, and higher values may hinder both the diffusion of coarse air bubbles and the movement of membrane fibres, in case hollow fibre (HF) membranes are used [10].

### 3.3. Dissolved Oxygen (DO)

Aeration in MBR systems is employed in order to: a) provide the necessary amount of oxygen to the biomass, b) effectively agitate the mixed liquor and keep it in suspension, and c) reduce fouling by scouring the membrane surface. Aeration influences indirectly fouling in MBRs, mainly by affecting the system’s biology, such as the bio-film structure, the SMP concentration level, the bio-floc size etc. [24]. In general, a DO concentration of 1–3 mg/L is sufficient, but effective agitation should be also applied in order to avoid the formation of ‘dead zones’, i.e., with lower oxygen concentrations, within the biomass. Higher DO concentrations generally increase membrane filterability, as indicated by the low specific resistance of cake layer on the membrane surface [25]. However, excessively high DO concentrations can significantly increase also the operational cost. On the other hand, lower DO concentrations (e.g., <1 mg/L) can increase fouling because larger amounts of EPS can be produced, due to microbial ‘stress’ under this condition.

Faust et al. [26] operated two MBR lab-scale units with different DO concentrations (1 and 4 mg/L) and observed that bio-flocs were larger, biomass filterability was higher and effluent quality was better for the higher DO concentration. In an effort to correlate the DO concentration with the composition of microbial communities, Gao et al. [27] applied three DO concentrations (0.5, 2 and 4 mg/L) in an MBR system, which treated municipal wastewater. Results showed that the reduction of DO concentration from 4 to 0.5 mg/L resulted in the production of more EPS, due to the growth and predominance of specific bacterial species. Ji and Zhou [28] also claimed that aeration directly controls the concentration and composition of EPS in the bio-flocs and the protein/carbohydrate ratio on the membrane surface. It has been also observed that, as the thickness of the bio-film layer increases with filtration, some bio-film regions become anaerobic and therefore, can increase fouling within a mostly aerobic bio-film [29]. Finally, the phenomenon of endogenous decay can also significantly increase the concentration of EPS, since the transition between aerobic to anaerobic conditions appears to increase the amount of EPS.

### 3.4. Foaming

Foaming can be described in this case as floating biomass and is generally attributed to the combination of surfactants (detergents) presence (of anthropogenic origin), of bio-surfactants formed by microorganisms, and/or of two specific groups of filamentous bacteria, i.e., *Gordonia* spp. (or *Nocardia* sp.) and *Microthrix parvicella*. Foam is regarded as a three-phase matrix, since it comprises gas (air bubbles), liquid (wastewater) and solid particles (the biomass containing hydrophobic filamentous bacteria). These microorganisms are not separated following the typical cell division process and tend to grow in the form of ‘filaments’. *Gordonia* spp., also known as *Actinomycetes*, are filamentous bacteria, which are extremely hydrophobic, due to the presence of mycolic acids on their cell walls. *Microthrix parvicella* is also hydrophobic, utilizes long-chain fatty acids as a carbon source and has an increased affinity for water-insoluble fats and lipids, due to their hydrophobicity. The mycolic acids in their cell walls make them sufficiently hydrophobic to become attached on the gas bubbles and rise to the surface of the liquid, resulting in the formation of rather stable foams.

Foaming is a well-studied problem by many researchers with significant impact on the process efficiency, especially in the activated sludge process, where the fundamental separation method is settling/sedimentation, and not filtration, as in the MBR technology. Several studies have demonstrated a clear correlation between foaming and the presence of surfactants, bio-surfactants and the mycolic acid-containing microorganisms. Other studies have showed that foaming is initiated primarily by the presence of surfactants and bio-surfactants. However, critical concentrations for foam initiation have not been yet determined, due to the numerous compounds which are involved, and their variability among sludges of different origin. Foam stabilization is mainly due to the filamentous *Gordonia* and *Microthrix parcivella* species, but there is evidence that other non-filamentous mycolic acid-containing microorganisms might also act as stabilizing agents. In MBRs, foaming appears to decrease the membrane permeability, due to the higher hydrophobicity of foaming sludge [10]. Recently, the study of Banti et al. [30] showed that the filamentous bacterial population can be appropriately manipulated through the adjustment of the MBR configuration, and effectively result in the mitigation of membrane fouling.

### 3.5. Hydrophobicity and Surface Charge

A number of research studies have shown that membrane fouling is favoured by the presence of highly hydrophobic bio-flocs, although the direct effect of hydrophobicity on fouling is difficult to be assessed. The EPS concentration and the filamentous index, i.e., a parameter describing the relative presence of filamentous bacteria in the biomass, directly influence the bio-floc hydrophobicity, as well as the respective surface charge (zeta-potential). The excess growth of filamentous bacteria has been reported to yield higher EPS concentrations, lower zeta-potentials, more irregular bio-floc shapes and higher biomass hydrophobicity. Surface charge and zeta-potential of bio-flocs (and of EPS) are usually in the range of 0.2–0.7 meq/g VSS and 20–30 mV respectively, and this is mostly attributed to the anionic nature of the respective functional groups of Natural Organic Matter (NOM) [10].

## 4. Mixed Liquor/Biomass Characterization

Various methods have been employed in order to assess the mixed liquor quality, in terms of filterability and membrane fouling potential. Ideally, the measurement of filterability can be indicative of upcoming accelerated fouling. The applied methods can be either direct, which are based on free drainage, vacuum drainage and cross-flow filtration, or indirect, which are based on the measurement/analysis of Capillary Suction Time (CST), colloidal TOC, particle size and Relative Hydrophobicity (RH). Although more than one of these methods may be useful for a given case study, none of them is universally reliable for all conditions. This is because the filtration mechanisms, which take place during the test methods, are significantly different from those encountered during the actual membrane filtration in the MBRs. Some of the mostly applied filterability test methods are presented in the following sections.

### 4.1. Direct Test Methods

#### 4.1.1. Free Drainage Test (FDT)

The Free Drainage Test (FDT) is the simplest employed method and does not require any special equipment. A filter paper (e.g., Whatman 42, i.e., pore size of 2.5 μm) is folded and placed in a funnel. After pouring 50 mL of mixed liquor into the funnel, the filtrate volume is measured after a predetermined time (e.g., 5 min).

Although the FDT is useful to obtain a rough idea on fouling potential, the following limitations appear [11]:The quality of the results can vary significantly because relatively small amounts of mixed liquor are used.There is only a slight driving force (i.e., gravity) to filter the mixed liquor, leading to the formation of a much milder, i.e., less compact, cake layer. Therefore, the obtained results are nearly free of the cake layer compaction effect.No cross-flow exists in the funnel and thus, all the particles and macromolecules in the sample can contribute to the cake layer formation. In contrast, larger particles are transported back to the bulk and do not contribute to cake layer formation in the actual filtration in MBR systems.Unlike the actual filtration in MBRs, where macromolecules (e.g., EPS) are the major components of the cake layer formation, in the FDT solid particles are the major components.Filterability is affected by the MLSS concentration, due to the lack of normalization against MLSS.Results are affected by the atmospheric pressure.

#### 4.1.2. Modified Free Drainage Test (MFDT), or Sludge Filterability Index (SFI)

In order to overcome the aforementioned limitations, the FDT should be appropriately modified. During the Modified Free Drainage Test (MFDT) the mixed liquor, which is poured in a Buchner funnel, can be mixed by a flat blade (Figure 4). A sample of 500 mL is heated or cooled to 20 °C in a water bath before it is poured into the funnel in order to reduce the temperature effect. Then, it is mixed at 40 rpm by a clamp blade agitator at a height of 1 mm above the filter paper. Filter papers with a diameter of 150 mm and a pore size of 0.6 μm has been employed. The time (Δt) which is required to obtain 100–150 mL of filtrate is measured. Finally, the Sludge Filterability Index (SFI) is calculated by normalizing the time against MLSS, as shown in Equation (2) [31]:(2)$$\mathrm{SFI}=\frac{\Delta t\;\left(s\right)}{\mathrm{MLSS}\;\left(\%\right)}$$

#### 4.1.3. Time to Filter (TTF_100_)

The Time-To-Filter (TTF_100_) method is a well-established method which can be used as an easy and relatively rapid way to assess the mixed liquor filterability. During the TTF_100_ method, a Buchner funnel with a diameter of 90 mm and Whatman #1, #2, i.e., pore size of 11 μm and 8 μm, respectively, or equivalent filter papers are used (Figure 5). After pouring 200 mL of mixed liquor on the Buchner funnel, the time which is required to obtain 100 mL of filtrate is measured at a vacuum pressure of 51 kPa (designated a TTF_100_) [32]. In some studies, except for the TTF_100_, the times which are required to obtain 20, 40, 60 and 80 mL of filtrate, i.e., TTF_20_, TTF_40_, TTF_60_ and TTF_80_ respectively, are also measured, in order to acquire an extended profile of recorded TTF times, which will contribute to a better comparison and understanding of the obtained results [33]. According to Côté [34], a TTF_100_ less than 100 s indicates high filterability of the mixed liquor, 100 to 200 s indicate intermediate filterability and more than 300 s indicate poor filterability.

#### 4.1.4. Modified Fouling Index (MFI)

The Modified Fouling Index (MFI) can be defined, assuming that the cake layer is the only cause of flux loss and its weight increases proportionally to the filtrate volume, applying a dead-end filtration mode. As it is known, the relationship between the TMP and the flux can be described by using a simple resistance-in-series model (Equation (3)). This equation is fundamentally the same with other equations, which are used to model heat and mass transfer, electrical conduction, air/water flow through pipelines etc., where the flux/flow/current is proportional to the driving force and inversely proportional to the respective resistance:(3)$$J=\frac{{\Delta P}_{T}}{\mu \;{(R}_{m}+R_{c}+R_{\mathrm{irr}})}$$
where J is the water flux (m/s), ΔP_T_ is the TMP (Pa or kg/m/s^2^), μ is the permeate viscosity (kg/m/s or cP, 1.00 ∙ 10^-3^ for water at 20 °C), R_m_ is the membrane resistance (m^−1^), R_c_ is the cake layer resistance or reversible fouling resistance (m^−1^) and R_irr_ is the irreversible fouling resistance (m^−1^).

Equation (4) is the modification of Equation (3), where irreversible fouling resistance R_irr_ is assumed zero and the flux J is written in a more detailed form. According to the initial assumption, the cake resistance R_c_ is proportional to the amount of the particles, which are deposited on the membrane surface. This amount is calculated by the particle concentration C (kg/m^3^), the filtrate volume V (m^3^) and the membrane surface area A (m^2^), as shown in Equation (5), where α (m/kg) is a proportional constant. By inserting Equation (5) to Equation (4), Equation (6) is obtained. Applying the boundary conditions of (0, 0) and (t, V), Equation (6) can be solved to give Equation (7). Finally, Equation (7) is simplified to Equation (8), where the MFI is defined as αμC/2ΔPA^2^.
(4)$$\frac{1}{A}\frac{\mathrm{dV}}{\mathrm{dt}}=\frac{\Delta P}{\mu \;{(R}_{m}+R_{c})}$$
(5)$$R_{c}=\frac{\mathrm{aCV}}{A}$$
(6)$$\int_{0}^{V}{(R}_{m}A+\mathrm{aCV})\;\mathrm{dV}=\frac{{\Delta \mathrm{PA}}^{2}}{\mu }\int_{0}^{t}\mathrm{dt}$$
(7)$$\frac{1}{V}=\frac{{\mu R}_{m}A}{{\Delta \mathrm{PA}}^{2}}+\frac{a\mu C}{{2\Delta \mathrm{PA}}^{2}}V$$
(8)$$\frac{1}{V}=\alpha +\mathrm{MFI}\;V$$

The MFI is measured using a stirred cell, applying a dead-end filtration mode at 207 kPa, and is denoted as either MFI_0.45_ or MFI_UF_, in order to remark the used filters, i.e., 0.45 μm filter or an appropriate ultrafilter, respectively. Typically, the filtrate volume is recorded more frequently than every 30 s with a timer. It must be stated that many experimental variations exist for the measurement of MFI in terms of the stirring speed, the membrane pore size, the pressure etc., depending on the purpose of the test and the specific sample characteristics, causing difficulties, regarding the data inter-comparisons between different researchers.

Simpler filterability test methods have been also proposed. Tarnacki et al. [35] estimated the relative fouling potential of mixed liquor by measuring the time, which is required to obtain a certain volume of filtrate under a constant pressure. In this experiment, a custom-made stirred cell and membranes with an area of 38 cm^2^ were used at 1 bar, 20 °C and 400 rpm. Filterability and fouling indices are obtained as follows:1)Filterability, L_15_ (L/m^2^∙h/bar), is estimated after a filtrate volume of 15 mL is obtained by dividing the flux by pressure. However, L_15_ cannot be normalized against a wide range of pressures, because flux is not proportional to the pressure, when the cake layer dominates the filterability decrease. Thus, the best practice to perform this filtration test is at a fixed ΔP (TMP) value Equation (9):
(9)$$L_{15}=\frac{J_{15}}{\Delta P}$$Fouling index, FI_30_, is calculated by dividing the observed flux after 30 min of filtration, J_30_, by the initial water flux, J_w,0_. Similarly to L_15_, FI_30_ is also affected by the TMP and therefore, it should be measured at a fixed TMP value Equation (10):(10)$${\mathrm{FI}}_{30}=\frac{J_{30}}{J_{w,0}}$$

Although several attempts were performed to effectively develop fouling indices, which reasonably represent the fouling potential in MBRs, only limited success has been accomplished. Most of the assumptions which are adopted, when developing the previous equations, do not match with reality, because of one or more of the following reasons:The weight of cake layer is not proportional to the filtrate volume in the actual membrane filtration, because of the existence of the particle back-transport phenomenon. Depending on the flux profile during filtration, not only the amount of the deposited particles, but also the structure of cake layer can vary.The cake layer compaction plays an important role in cake resistance, but MFI equations do not consider it. Depending on the pressure and flux during the stirred cell test, the extent of cake layer compaction may vary.Whichever laboratory test method is used, the hydrodynamics on the membrane surface are not the same as in real membrane filtration. Hydrodynamic conditions determine particle back-transport velocity and eventually the membrane fouling rates.The MFI is measured under a membrane fouling condition typically at higher TMP values, than in actual filtration. Cake layer formation and ageing patterns in short-term filtration procedures cannot be the same as in long-term filtration.

#### 4.1.5. Delft Filtration Characterization Method (DFCM)

The Delft Filtration Characterization Method (DFCm) employs a single tubular ultrafiltration membrane with an internal diameter of 8 mm and a nominal pore size of 0.03 μm under inside-out filtration mode. Cross-flow velocity is maintained at 1 m/s by using a peristaltic pump and flux is maintained at 80 L/m^2^h. During this test, several characteristics are recorded, such as pressure (at the feed concentrate and at the permeate), mixed liquor temperature, pH, DO concentration and flux. Compared to other filterability test methods, the DFCM provides a closer representation of the real membrane filtration, but it is still not exactly the same. Nonetheless, the applied flux (80 L/m^2^∙h) is much higher than the typical fluxes encountered in most of submerged MBR processes (15–25 L/m^2^∙h). The high flux is inevitable in order to facilitate membrane fouling and complete the experiment within a reasonable time; however, it results in the overcontribution of larger size particles to the filtration resistance. In fact, at high fluxes larger particles can deposit on the membrane surface more easily, than at lower fluxes. In addition, the cake layer compaction is more significant at higher fluxes and this is why this test mainly differs from the actual membrane filtration in MBRs. The higher shear stress in the circulation pump can also disrupt the bio-floc particles and make them smaller in DFCM [11].

### 4.2. Indirect Test Methods

#### 4.2.1. Capillary Suction Time (CST)

The Capillary Suction Time (CST) method has been used to measure the dewatering properties of biomass. When the biomass comes in contact with filter paper, the contained water starts to wet the paper due to capillary suction phenomena and the biomass becomes more compact. The dense biomass layer, which is formed on the interface, acts as a barrier for further water loss to the filter paper and therefore, the filterability of biomass in the biomass-paper interface determines the CST. If macromolecules and fine particles are abundant in the biomass, more compact biomass layers can be formed in the interface, and then the wetting speed decreases and the CST increases. According to APHA [32], 10 mL of mixed liquor is poured into the test cell (with an internal diameter of 18 mm and a height of 25 mm). As soon as the mixed liquor contacts the filter paper, it starts to wet it and proceeds radially. The time which is required for the water to proceed from the initial radius r_1_ = 15.9 mm to the final radius r_2_ = 22.2 mm is measured by specially designed conductivity sensors and is called CST. CST values can be further normalized by dividing them by the MLSS concentration (g/L), but this normalization is effective only for a narrow MLSS range [11].

#### 4.2.2. Colloidal TOC Measurement

The concentration of colloids in the mixed liquor can be used as a membrane fouling index. The colloidal TOC is defined as the difference between the filtrate TOC of a 1.5 μm-filter paper (934-AH, Whatman, Piscataway, NJ, USA) and the permeate TOC of a MBR. Colloidal TOC has been known as a good indicator of membrane fouling potential in MBR systems in several different locations. In a pilot MBR which treated real municipal wastewater [36], it was observed that the critical flux was inversely correlated with the colloidal TOC. In the same study, TTF_100_, MLSS and permeate TOC appeared to be less correlated with the flux, than the colloidal TOC. It has been suggested that a colloidal TOC of less than 10 mg/L indicates a mixed liquor of “good” condition, 10–20 mg/L an intermediate condition and more than 20 mg/L a marginal condition in terms of filterability.

#### 4.2.3. Particle Size Analysis (PSA)

Fouling rate increases as the number of submicron particles increases in the membrane tank of MBR systems. This is due to the slow particle back-transport velocity of these particles and the strong interaction with the membrane surface through surface charge and van-der-Waals forces. Although the instruments, which are based on laser light scattering, are not accurate in measuring such small particles, the shifting of the particle size distribution (PSD) curve to smaller sizes can be deemed as an increase of submicron particles. In a side-by-side test which employed a lab-scale MBR with side-stream tubular membranes [37], fouling rate increased as the average particle size measured by the particle size analyser decreased. Deflocculation and accelerated membrane fouling can be also caused by undesirable biological-related conditions, e.g., by low DO concentration, by high F/M ratio, or by low SRT values. However, it must be noted that although the decrease of average particle size accelerates membrane fouling, small average particle sizes, which remain at relatively constant level, is not considered necessarily as a problem [11].

#### 4.2.4. Relative Hydrophobicity (RH) Measurement

Relative hydrophobicity (RH) has been used in the field of biotechnology as an index of the physiological state of microorganisms. RH is measured by a test called “Microbial Adhesion To Hydrocarbons” (MATH). The MATH test is performed with organic solvents, such as *n*-hexane, *n*-octane, *n*-octanol, *n*-dodecane etc. A mixed liquor sample and a solvent are poured into a funnel which is then shaked and the two phases (aqueous/organic) are separated. Finally, the ratio of microorganisms in both phases is measured based on light absorbance.

In the original version of MATH test, a sample of 4 mL is mixed with 1 mL of *n*-dodecane and vortexed for 2 min, followed by resting for 15 min in order to enable the respective phases’ separation. After taking 0.75 mL of aqueous sample, the absorbance (S_e_) is measured at 400–600 nm. The absorbance of the original aqueous sample (S_i_) is also measured at the same wavelength. Finally, RH is calculated using the following Equation (11) [38]:(11)$$\mathrm{RH}\;(\%)=\left(1-\frac{S_{e}}{S_{i}}\right)\times 100$$

Because membranes are prone to fouling by the hydrophobic components of the mixed liquor through hydrophobic-hydrophobic interactions, the RH can indirectly be used as a membrane fouling index. There are many variations of the MATH test, depending on the solvent, the wavelength, the vortexing method and its duration etc. In some cases, the amount of dissolved matter in the solvent phase is used as an RH index after the evaporation of solvent. RH is an indicative parameter for membrane fouling, but it alone cannot be used to assess fouling propensity, as it highly depends on other biomass characteristics as well [11].

## 5. Conclusions

The biomass characteristics which mostly affect membrane fouling in MBR systems include the concentration of MLSS, colloidal matter and EPS. Although initially regarded as major foulants, MLSS content is now considered to be only weakly correlated with membrane fouling, especially within moderate concentration ranges (e.g., 3.5–8.5 g/L). However, excessively high values (>20–30 g/L) should be avoided, because they can cause accelerated fouling through the increase of viscosity, which hinders air diffusion (aeration) in the mixed liquor. Colloids, which are formed during the microbial metabolism or derive from the influent substrate, can contribute to fouling; however, more recent studies consider EPS to be the most significant parameter, especially when they are in soluble form (i.e., as sEPS or SMP) and not directly associated with the microbial cell (i.e., as bEPS). SMP can be adsorbed onto the membrane surface, block the membranes pores and/or form a gel layer. Their carbohydrate fraction (SMP_c_) is considered to have the most important role in fouling, mainly due to the hydrophilic nature and gelling properties, which are exhibited by the polysaccharides. Other fouling-related wastewater and biomass properties may include viscosity, temperature, DO, foaming, hydrophobicity and surface charge. However, these properties can affect membrane fouling indirectly, through the modifications they induce to the aforementioned major biomass characteristics (MLSS, colloids and EPS) and their role should not be underrated. Finally, the applied methods for biomass characterization, in terms of filterability assessment and fouling potential, can be both direct, which are based on free drainage, vacuum drainage and cross-flow filtration, or indirect, which are based on the measurement/analysis of CST, TOC, particle size analysis and relative hydrophobicity.

## Figures and Tables

**Figure 1 molecules-24-02867-f001:**
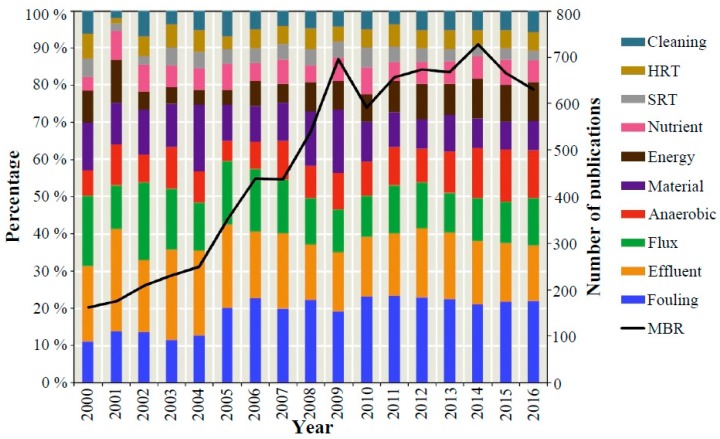
Research trends in MBRs based on the publication numbers in each key subject area in the Scopus (abstract and citation database of peer reviewed literature) [2].

**Figure 2 molecules-24-02867-f002:**
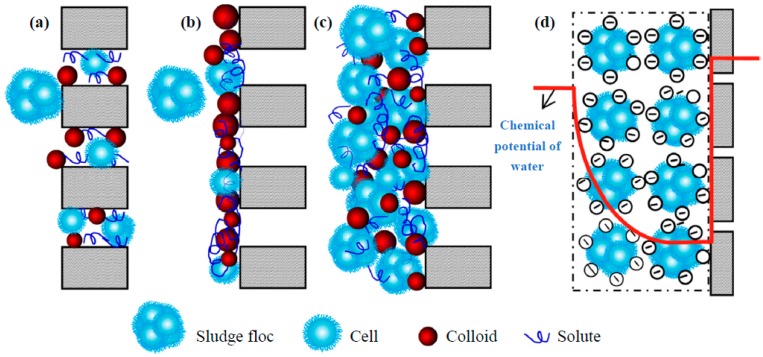
Schematic illustration of some membrane fouling mechanisms: (**a**) pore clogging, (**b**) gel layer formation, (**c**) cake layer formation, and (**d**) osmotic pressure effect [3].

**Figure 3 molecules-24-02867-f003:**
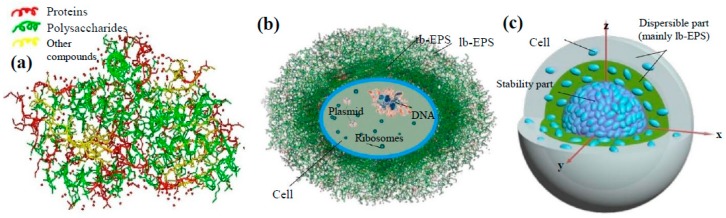
Schematic of: (**a**) EPS structure, (**b**) cell structure and (**c**) bio-floc structure [3].

**Figure 4 molecules-24-02867-f004:**
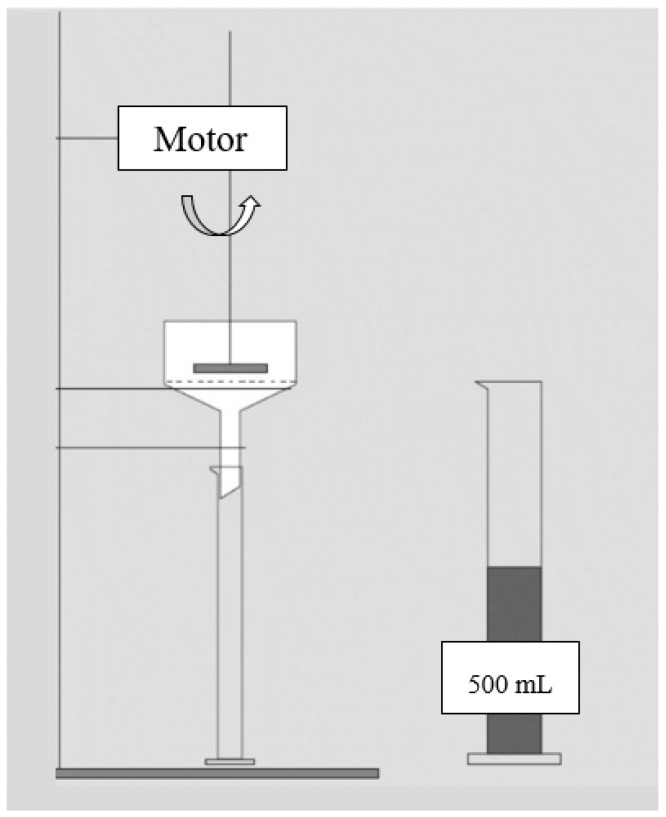
Schematic of the apparatus for measuring the SFI [11].

**Figure 5 molecules-24-02867-f005:**
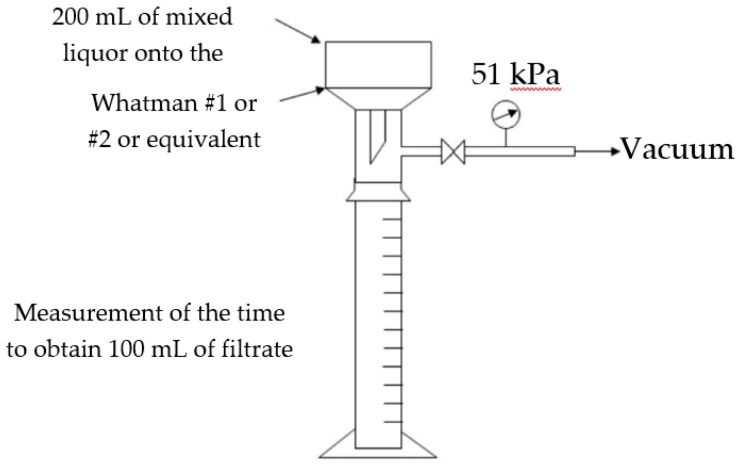
Schematic of the apparatus for measuring the TTF_100_ [11].

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
