# Peer review of "Biomass Characteristics and Their Effect on Membrane Bioreactor Fouling"

_molecules, 2019, doi:10.3390/molecules24162867_

Round 1
Reviewer 1 Report
The manuscript entitled “Biomass characteristics and their effect on MBR fouling” offers a good contribution for understanding the effect of biomass characteristics on membrane fouling in MBR, In addition, it gives sufficient description of methods used for characterization and assessment of biomass quality. I would recommend publication in "Molecules" after some minor edits for improving the manuscript. The remarks are provided as follow:
1) the authors should improve the resolution of the abscissa scale of Figure 1
2) please add references for the sections “3.1 Temperature” and “3.2 Viscosity”,
3) line 455, regarding the Whatman grade 42, I suggest adding the pore size of the filter
4) In figure 3:
change “ml” to “mL”,
please specify the meaning of “M” and "46 cm" shown in the figure
5) lines 355, 474, 507 and 532 change to “°C”
6) figure 4 please remove the sign in the word “Time” regarding the sentence “measurement of the time to obtain 100 mL of filtrate”
7) for Whatman filter paper (Grade 1 and 2), I suggest adding the pore size of the filter used for the test
Reviewer 2 Report
This manuscript reviewed biomass characteristics and their effect on MBR fouling. The key characteristics, factors affecting these characteristics, and characterization methods were summarized in this manuscript. Although the review is not comprehensive, it gives some interesting information, that adds the literature. However, there are still some issues should be addressed before its acceptance of potential publication.
1). The graphical abstract or cover art should be provided.
2). The abstract is more like background introduction rather than summarization of review content. These background can be put in “introduction” section, and review content should be presented.
3). The authors just copy the figures from the Meng and Lin's review articles. This is not allowed for most of journals.
4). The title of Section 2.1.3 is a interrogative sentence, which is not a professional way.
5). In "introduction" section, the authors didn't mention membrane fouling mechanisms. I think it is unacceptable for a review. I suggest to add a separated paragraph introducing the primary fouling mechanisms. The primary fouling mechanisms including adhesion/deposition mechanism and filtration resistance caused by chemical potential mechanism have not been introduced. As for adhesion/deposition process, interfacial interactions (thermodynamic forces) between foulants and membrane are decisive forces. Several important references (such as Water Res. 2019,149: 477-487) can be cited to provide a comprehensive background. It has been recently revealed that chemical potential mechanism related with foulant/cake layer filtration is mainly responsible for the filtration resistance. These literature studies should be briefly introduced to deepen this study.
6). I suggest to add a figure illustrating membrane fouling mechanisms. The idea of this figure can refer to Journal of Membrane Science, 2014, 460: 110-125.
7). The length of section “4.Mixed liquor/biomass characterization” can be significantly reduced since it is weakly correlated with fouling issue.
Round 2
Reviewer 2 Report
Since the authors have well addressed all of my concerns, I suggest publication of this work in this journal.